# Species Surface Distribution and Surface Tension of Aqueous Solutions of MIBC and NaCl Using Molecular Dynamics Simulations

**DOI:** 10.3390/polym14101967

**Published:** 2022-05-12

**Authors:** Omar Alvarado, Gonzalo R. Quezada, Jorge H. Saavedra, Roberto E. Rozas, Pedro G. Toledo

**Affiliations:** 1Departamento de Química, Facultad de Ciencias, Universidad del Bío-Bío, Av. Collao 1202, Concepción 4030000, Chile; oalvarado@ubiobio.cl; 2Departamento de Ingeniería Química, Universidad de Concepción, Concepción 4030000, Chile; 3Department of Wood Engineering, Universidad del Bío-Bío, Av. Collao 1202, Concepción 4030000, Chile; jsaavedra@ubiobio.cl; 4Department of Physics, Universidad del Bío-Bío, Av. Collao 1202, Concepción 4030000, Chile; rrozas@ubiobio.cl; 5Department of Chemical Engineering and Laboratory of Surface Analysis (ASIF), Universidad de -Concepción, Concepción 4030000, Chile

**Keywords:** MIBC, NaCl solution, surface adsorption, density profiles, air–liquid interface structure, surface tension, polarizable fields, molecular dynamics simulation

## Abstract

Methyl isobutyl carbinol (MIBC) is a high-performance surfactant with unusual interfacial properties much appreciated in industrial applications, particularly in mineral flotation. In this study, the structure of air–liquid interfaces of aqueous solutions of MIBC-NaCl is determined by using molecular dynamics simulations employing polarizable and nonpolarizable force fields. Density profiles at the interfaces and surface tension for a wide range of MIBC concentrations reveal the key role of polarizability in determining the surface solvation of Cl^−^ ions and the expulsion of non-polarizable Na^+^ ions from the interface to the liquid bulk, in agreement with spectroscopic experiments. The orientation of MIBC molecules at the water liquid–vapor interface changes as the concentration of MIBC increases, from parallel to the interface to perpendicular, leading to a well-packed monolayer. Surface tension curves of fresh water and aqueous NaCl solutions in the presence of MIBC intersect at a reproducible surfactant concentration for a wide range of salt concentrations. The simulation results for a 1 M NaCl aqueous solution with polarizable water and ions closely capture the MIBC concentration at the intercept. The increase in surface tension of the aqueous MIBC/NaCl mixture below the concentration of MIBC at the intersection seems to originate in a disturbance of the interfacial hydrogen bonding structure of the surface liquid water caused by Na^+^ ions acting at a distance and not by its presence on the interface.

## 1. Introduction

A large number of industrial processes are based on the properties of liquid–fluid interfaces loaded with surface-active agents. The surfactants used have molecular architectures of varying complexity. MIBC (methyl isobutyl carbinol) is a high-performance surfactant whose unusual interfacial properties are commonly required in industrial applications, particularly in mineral flotation. Over the years, abundant knowledge has accumulated from both field and laboratory on the application of MIBC in water [1,2]. However, without a complete knowledge of the behavior of air–solution interfaces in the presence of MIBC, the critical shortage of water in some mining regions has forced the use of seawater as it is at least partially desalinated. 

The ionic structure of the air–water interface has been extensively studied over the years. The air–solution interface for several salts has received significant attention in the last two decades for its key role in environmental chemistry and atmospheric science through molecular simulation. However, the ionic structure of the air–solution interface in the presence of a surfactant is far from fully understood. In particular, the air-NaCl solution system has been studied for wide ranges of MIBC concentration by molecular simulation with nonpolarizable force fields [3,4,5]. However, these fields repel the Na^+^ and Cl^−^ ions from the interface, which is fine for Na^+^ but not for Cl^−^, for which the surface tension cannot be predicted correctly. The existence of chloride ions and other halides at several sodium halide air–solution interfaces was demonstrated first by molecular dynamics simulations of polarizable ions [6,7,8,9] and then by direct experimental measurement with spectroscopic selective interface techniques [6,7,8,9,10,11,12,13,14,15,16], contradicting the predictions of the classical Onsager–Samaras model [17] that indicate that sodium cations and halide anions are repelled from the interface. Currently, theory and experiments have set up a picture in which the heavier, and therefore more polarizable, halide anions reside in the interface rather than inside the bulk solution. The structure of the ions adsorbed at the liquid–gas interfaces determines the macroscopic surface tension which decreases in the presence of surfactants, such as MIBC, and increases in the presence of inorganic electrolytes such as NaCl. The extent of the change in surface tension depends on the nature and concentration of the species at the interface. 

Previously, in 2009, Ozdemir et al. recognized that the surface tension curves of water and an aqueous NaCl solution intersect at one concentration of MIBC: ca. 0.7 mM at 296 K [18]. More recently, Phan et al. (2012), Shahir et al. (2016) and Gao et al. (2017) found experimentally that the value is ca. 1 mM [5,19,20]. In 2013, Castro et al. found for the aqueous NaCl/MIBC mixture that the concentration of intersection is ca. 1.2 mM at 300 K and that this concentration has universal character because it is reproduced in NaCl solutions with a wide range of salt concentrations [21]. Castro et al. considered this concentration as surface tension switch point (s.t.s.p.). Below this critical concentration, the surface tension is higher than in pure water because the ions of the NaCl salt apparently dominate the liquid–vapor interface, increasing its rigidity. above the critical concentration of MIBC, the surface tension lowers because MIBC molecules occupy the interface, displace the other salt ions, and display their surface-active power. This behavior has been observed in several other solutions of alkaline salts, MIBC [18] and other non-ionic surfactants [18,22,23,24,25], each with its own critical concentration.

In this paper, the structure of air–liquid interfaces of aqueous solutions of MIBC-NaCl is determined using molecular dynamics simulations. Na^+^, Cl^−^ and MIBC density profiles are compared for simulations employing polarizable and nonpolarizable force fields. An attempt is made to show the key role of polarization of water and ions in the surface solvation of Cl^−^ ions and in the expulsion of non-polarizable Na^+^ ions from the interface to the bulk. The surface tension is also calculated; the interest is to capture the intersection of the surface tension curves of MIBC in water and in 1 M NaCl solution. Simulation results with polarizable ions and water compare well with available experimental data. These results are expected to have implications for the optimization of flotation operations and for several other chemical processes.

## 2. Methodology

Interfacial properties of MIBC in aqueous solutions of 1 M NaCl were studied by means of molecular dynamics simulations performed using GROMACS 2016-dev-20170105-c53d212-dirty version available on gromacs.org, where Lagrangian dynamics for polarizable systems [26] are implemented. Three systems were considered. The System A non-polar, the SPC/E water model to describe water., the Lennard-Jones 12-6 parameters from Li et al. [27] were used to describe ion–oxygen distances (IOD) for the SPC/E water model, and the AMBER99sb force field [28,29] was used to describe the MIBC molecules (Table 1). System B was the same as System A, except that the Drude 2013 force field [30] was used to describe polarizable ions. System C was the same as B, except that the SWM4-NDP water model [31] was used to describe polarizable water. MIBC polarization was not included. Lorentz–Berthelot mixing rules were employed for the crossed interactions between atoms. The systems are summarized in Table 2. The highest frequencies of our systems make possible the use of 1 fs without compromising the stability of the velocity Verlet method. The particle mesh Ewald method (PME) was used for long-range corrections of the electrostatic potential [32]. The van der Waals and Coulombic cutoff radii were 1.2 nm. Temperature was controlled with the Berendsen thermostat for the non-polar system (A) with a relaxing time of 0.1 ps and with Nosé–Hoover thermostat for the polar systems (B and C) with a relaxing time of 0.1 ps for the atoms and 0.005 ps for the Drude particles. The reference temperature was 300 K for atoms and 1 K for Drude particles. For the polarizable system, the Drude mode was Lagrangian with a hard-wall constraint and a constraint length of 0.02 nm. 

The simulated system was a film of liquid water surrounded by its vapor. The system was prepared in different simulation steps. First, Na^+^ and Cl^−^ ions were disposed homogeneously in a crystalline arrangement in a simulation box of lengths $L_{x}=L_{y}=3$ nm and $L_{z}=4$ nm with a concentration of 1 M. Then water molecules were added to the system, avoiding overlaps. Then the simulation box length *z* was expanded to 10 nm in one step so that two liquid–vapor interfaces perpendicular to the *z*-axis were created (see Figure 1). Then, MIBC molecules were added at positions close to the interface. This initial configuration was relaxed in a force-minimization run using the steepest descent algorithm and then equilibrated in an NVT simulation at 300 K for 1 ns (Figure 1). During the simulations, periodic boundary conditions were applied in all directions. The production step consisted of 10 ns in an NVT ensemble. The data was collected every 1 ps. The density profiles of the two liquid–vapor interfaces were obtained and averaged. The reference plane of the liquid–vapor interface, *z* = 0, was defined as the position at which the density is the average between the liquid and vapor phases. From these data, averaged density profiles along the interface were obtained. The position of the interface, *z* = 0, was defined as the point where the density of water equals its mean value along the liquid–vapor interface. 

## 3. Results

### 3.1. MIBC Density Profiles and Orientation

For the three systems of interest, Figure 2 shows MIBC density profiles in the vicinity of the liquid–vapor interface for water and 1.0 M NaCl solution. For the calculations, the center group of MIBC is considered. Clearly, MIBC molecules prefer the interface not only in pure water but also in the presence of NaCl salt ions. As the concentration of MIBC increases, its density at the interface increases as expected, regardless of the system with/without salt and with/without polarization. What is new is that increases in MIBC concentration with/without NaCl also lead to increasing displacements of the MIBC molecules towards the water vapor phase, although the inclusion of water and ion polarizability leads to a more symmetrical distribution of MIBC molecules in the liquid–vapor interface. It is interesting to note in Figure 2 that the density profiles change more radically when considering the polarization of ions only, especially at the highest concentrations of MIBC with/without NaCl. The differences are minor when considering the polarization of water in addition to that of ions. However, it is clear that the interaction of polarizable water with MIBC molecules at the interface is stronger.

It is of great interest to know the location and orientation of MIBC molecules in the vicinity of the water liquid–vapor interface. Thus, the position of the constituent atoms O, C1, C2 and C3 of each of the MIBC molecules (Figure 3a) is determined in each simulation. Calculations involved in the determination of RDF for each constituent atom show the positions of each MIBC atom that correspond to the highest frequencies in the RDF. Simulations with 2, 10, 20, 30 and 40 MIBC molecules are considered. The results in Figure 3b for water and in Figure 3c for 1 M NaCl aqueous solution show that there is no significant difference, because MIBC molecules dominate the interface and other ions are expelled. Nor is there much difference between System A without polarizable components and System B with polarizable ions; when the number of molecules considered is very low, such as 2 and 10, the MIBC molecules are distributed with their oxygen atoms well submerged in the liquid phase to the point that the carbon atoms are dragged very close to the interface by the side of the vapor phase. However, as the number of MIBC molecules increases, they are pushed into the vapor phase. In the extreme case of simulations with 40 MIBC molecules, the oxygen atoms are located on average very close to the interface but on the side of the vapor, and the corresponding carbon atoms are deep in the vapor. However, of greater interest are the results for System C with polarizable ions and water. Figure 4a,b shows that regardless of the number of molecules considered in each simulation, the MIBC oxygen atoms remain on average in the liquid phase, with the carbon atoms to their sides but in the vapor phase. The results also suggest that in brine, the MIBC molecules are pushed from the liquid phase a little more than in pure water, which means that salt ions also claim the interface. The orientation of the MIBC molecules at the water liquid–vapor interface changes when the concentration of MIBC increases. Low-concentration MIBC molecules tend to lie parallel to the interface. However, high-concentration MIBC molecules adopt orientations perpendicular to the interface with their oxygens as the only atoms submerged in the liquid phase. This behavior is observed in pure water and in 1 M NaCl solution. It can be deduced that the adsorption of the MIBC molecules orthogonal to the water liquid–vapor interface is due to the competition for space with other MIBC molecules and not to a polar–apolar repulsion with water. The result is a well-packed monolayer of MIBC.

### 3.2. Ion Density Profiles

Axial density profiles for Na^+^ and Cl^−^ ions at the water liquid–vapor interface for 1 M NaCl solutions are shown in Figure 5 for the three systems of interest. The impact of polarization is quite dramatic. For the non-polar system, System A, it is observed that both Na^+^ and Cl^−^ are repelled from the interface; they prefer the bulk liquid phase for all MIBC concentrations. Adding polarizability to the ions, as in System B, produces important changes. Na^+^ and especially Cl^−^ ions compete at the interface to the MIBC molecules, coexisting with MIBC at low concentrations and being expelled to the liquid phase at high concentrations of MIBC. When ions and water are polarizable, a few more Na^+^ and Cl^−^ ions prefer the interface, Cl^−^ much more than Na^+^. Under these conditions, the competition for the interface is somewhat stronger. Na^+^ and Cl^−^ coexist with MIBC even at high concentrations of MIBC. The preference of Cl^−^ ions for the interface is due to its polarizability. In System C, both Na^+^ and Cl^−^ seem to preferentially concentrate at the liquid–vapor interface, however, the density of Na^+^ and Cl^−^ in the bulk liquid solution is ca. 0.6 nm^−3^. Therefore, at the highest MIBC content used in the simulations, Na^+^ at the interface is as concentrated as in the bulk or slightly less and Cl^−^ is slightly more concentrated than in the bulk. This result is in agreement with molecular simulations employing polarizable force fields [6,7,9]. This slightly improved concentration of Cl^−^ at the liquid–vapor interface is not sufficient to significantly modify the interfacial water structure as do the heavier halogen anions but demonstrates that the propensity of the halogen ions for the interface correlates with polarization and ion size, as has been indirectly verified by spectroscopic experiments [8,9]. In parallel work not yet published, the surface tension curve vs. MIBC concentration in aqueous solutions of 1 M of KCl and 1 M of KI has been compared, demonstrating that in a wide range of MIBC, the surface tension of KCl solutions is higher than that of KI solutions. I^−^ is much more polarizable than Cl^−^, so it is effectively installed at the interface as if it were a surfactant and acting as such.

### 3.3. Water Orientation 

The effect of NaCl ions and MIBC concentration on the orientation of water molecules at the water liquid–vapor interface is analyzed. Orientation is given by $\cos\left(\phi \right)$, where ϕ is the angle between the vector opposite to the water dipole and the normal to the mineral interface (see definition in Figure 6a). Values of −1 indicate that the water dipole points away from the interface, i.e., water hydrogens point towards the liquid water bulk, and the opposite occurs when the value is 1. Values between −1 and +1 indicate partial orientation of the water molecules. Figure 6b,c summarizes the average of $\cos\left(\phi \right)$ as a function of the distance from the liquid–vapor interface.

In System A, without polarizable components and without salt and without MIBC, it is observed that the water molecules in the liquid phase under the interface tend to be oriented on average with their hydrogens pointing towards the liquid phase and with the oxygen atoms pointing towards the vapor phase ($\cos\left(\phi \right)<0$). Remarkably just in the plane that defines the interface, the molecules recover the typical characteristic of bulk water without any structure ($\cos\left(\phi \right)=0$). Finally, the water molecules in the vapor layer on the interface tend to be oriented on average with their hydrogens pointing towards the vapor phase and the oxygen atoms pointing towards the interface ($\cos\left(\phi \right)>0$). In this way, the oxygen atoms of the interfacial water on average always point to the interface. In the presence of MIBC, in the same System A, the dipoles of the interfacial water molecules are forced to point more directly to the vapor phase ($\cos\left(\phi \right)$ is less than in water without MIBC), so that the oxygen atoms of water can participate more effectively in the formation of hydrogen bonds with the polar groups OH of the MIBC molecules (see Figure 4). The alignment effect on interfacial water molecules increases with the concentration of MIBC affecting not only the water molecules in the liquid layer under the interface but also those in the vapor layer over the interface. The bulk water condition is fully recovered in the vapor phase, although far from the interface at the highest concentration of MIBC. System B, with polarizable ions, is not very different from the non-polar system, only that at the highest concentrations of MIBC, the alignment of the water molecules in the direction of the OH groups of the MIBC increases and extends a little more within the vapor phase. This alignment effect of MIBC increases significantly in System C with water and polarizable ions; the effectiveness in the formation of hydrogen bonds between water and MIBC is the highest in accordance with a higher adsorption of MIBC at the interfacial liquid water layer (Figure 2 and Figure 3), which suggests greater effectiveness as a surfactant at the highest concentrations of MIBC. 

In the presence of NaCl, large differences in the orientation of water molecules are observed. Clearly, the presence of ions and their ability to align water molecules in their hydration layers generate this difference with respect to results without salts. In non-polar System A, the average alignment effects of NaCl ions and MIBC molecules overlap to align water dipoles in the direction of steam more effectively. In System B, with polarizable ions, the average alignment effect of NaCl ions is in contrast to the alignment effect of MIBC and exceeds it at low concentrations of MIBC ($\cos\left(\phi \right)>0$). The result is a poor adsorption of MIBC mainly on the vapor side, suggesting low surface activity (see Figure 2 and Figure 3). In System C, the competition to align the interfacial water molecules to the ions and to the MIBC molecules favors the NaCl ions at a low concentration of MIBC, and the MIBC at the highest concentrations of MIBC.

### 3.4. Surface Tension

Predicting the surface tension of NaCl solutions is important in its own right but could also validate the molecular simulation results when polarization is included. Castro et al. found that for the NaCl/MIBC mixture, the concentration for the surface tension switch point (s.t.s.p.) is ca. 1.2 mM at 300 K and that this concentration has a universal character because it is reproduced by NaCl solutions with a wide range of salt concentrations. Below this critical concentration, the surface tension is higher than in pure water, and above the critical concentration, the surface tension lowers because MIBC molecules occupy the interface, displaces the other salt ions, and displays their surface-active power. In this work, the purpose is to capture the intersection of the surface tension curves of MIBC in water and in 1 M NaCl solution by molecular simulation. This has been tried before without including polarization, however, the data available are few and scarce exactly in the zone of intersection of the curves, apart from the fact that the surface tension is sub-predicted [5]. Our own attempts without polarization do not show intersection of the curves, and if they did, it would be at a concentration much higher than that observed experimentally.

Surface tension curves of water and 1 M NaCl solution for a wide range of MIBC concentrations, 2 to 40 molecules, or 0.092 to 1.85 mM, are shown in Figure 7, at 300 K, for the three systems of interest (Table 2). The results show for System A, without polarization, that the surface tension slowly decays with the concentration of MIBC, and only when the concentration is very high, 30 and 40 molecules of MIBC, the tension drops quickly, reflecting the surface-active condition of MIBC. However, the curves in water and in 1 M solution of NaCl never cross; the curve for the salt solution always remains above the curve in pure water, indicating that NaCl ions coexist with MIBC at the interface. For System B, with polarizable ions, although surface tension decreases faster in saline solution than in pure water, at higher concentrations of MIBC, the curves do not intersect, as experimental curves do, and if they did it would be at a concentration of MIBC higher than the critical concentration. Finally, the simulation results for System C, with water and polarizable ions, show that the intersection of the two surface tension curves has been captured, at ca. 25 molecules of MIBC, ca. 1.15 mM, almost the exact experimental value [5,19,21]. The surface tension at the highest concentrations of MIBC is lower than in salt-free water, which unraveled a cooperative mechanism of dissolved salt ions to increase the surface-active capacity of MIBC, because competing for the interface forces the surfactant to be optimally packaged with each molecule as orthogonally as possible with respect to the interface. Additionally, in System C, the surface tension of neat water at 300 K compares well with the values published for the SWM4-NDP water model (67 ± 4 dyn/cm) [31,33] and against experimental data (71.73 dyn/cm at 300 K [34]) and significantly improves the prediction of the SPC/E model (63.6 dyn/cm at 300 K [34]; 63.7 dyn/cm at 298.15 K [19]; 55.4 dyn/cm at 300 K [35]; 61.3 dyn/cm at 300 K [36]), another benefit of using polarization. The increase in surface tension of the aqueous MIBC/NaCl mixture below the critical MIBC concentration has more to do with the alignment of interfacial water molecules induced “at a distance” by NaCl, which stiffens the liquid surface, than by the very presence of NaCl at the interface.

## 4. Conclusions

The structure of air–liquid interfaces of aqueous solutions of MIBC-NaCl obtained by molecular dynamics simulations approaches the spectroscopic experiment only if polarizability is included. The main result, which is not reproduced by simulations employing nonpolarizable force fields, are the surface solvation of Cl^−^ ions and the expulsion of non-polarizable Na^+^ ions from the interface to the bulk. The results with polarizability also suggest that in brines, MIBC molecules are pushed from the liquid phase to the interface only slightly more than in pure water, which means that salt ions, particularly Cl^−^, compete strongly for the interface. The orientation of MIBC molecules at the water liquid–vapor interface changes as the concentration of MIBC increases, from parallel to the interface to perpendicular, with their oxygens submerged in the liquid phase and forming a well-packed monolayer. The surface tension at the highest concentrations of MIBC is lower than in salt-free water, unraveling a cooperative mechanism of dissolved salt ions to increase the surface-active power of MIBC, because competing for the interface induces the surfactant to be optimally packaged with each molecule as orthogonal as possible with respect to the interface. Simulation results with polarizable water and ions show that the concentration of MIBC at the intersection of the surface tension curves of MIBC in water and in 1 M NaCl solution compare precisely with experimental results. The increase in surface tension of the aqueous MIBC/NaCl mixture below the critical MIBC concentration has more to do with the alignment of interfacial water molecules induced “at a distance” by Na^+^, which stiffens the liquid surface, than by the very presence of Na^+^ at the interface.

## Figures and Tables

**Figure 1 polymers-14-01967-f001:**
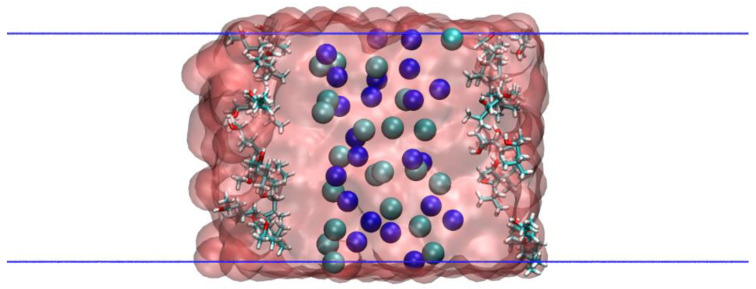
Initial configuration of a system with 1 M NaCl and 20 MIBC molecules. Na and Cl ions are represented respectively by blue and green spheres. MIBC is concentrated at the left and right interfaces.

**Figure 2 polymers-14-01967-f002:**
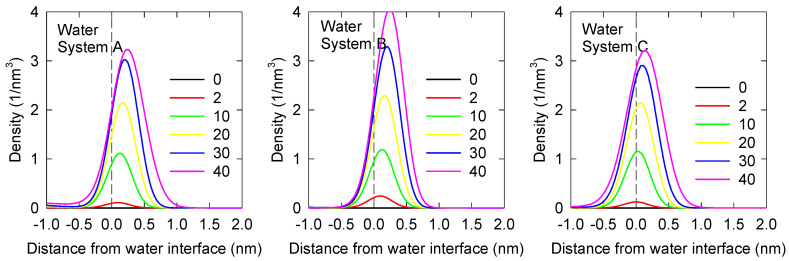
Axial density profiles of MIBC molecules in the vicinity of water liquid–vapor interfaces as a function of the concentration number of MIBC molecules (from 0 to 40) for water and 1.0 M NaCl solution for the three systems of interest (Table 2). Negative distances correspond to the liquid phase.

**Figure 3 polymers-14-01967-f003:**
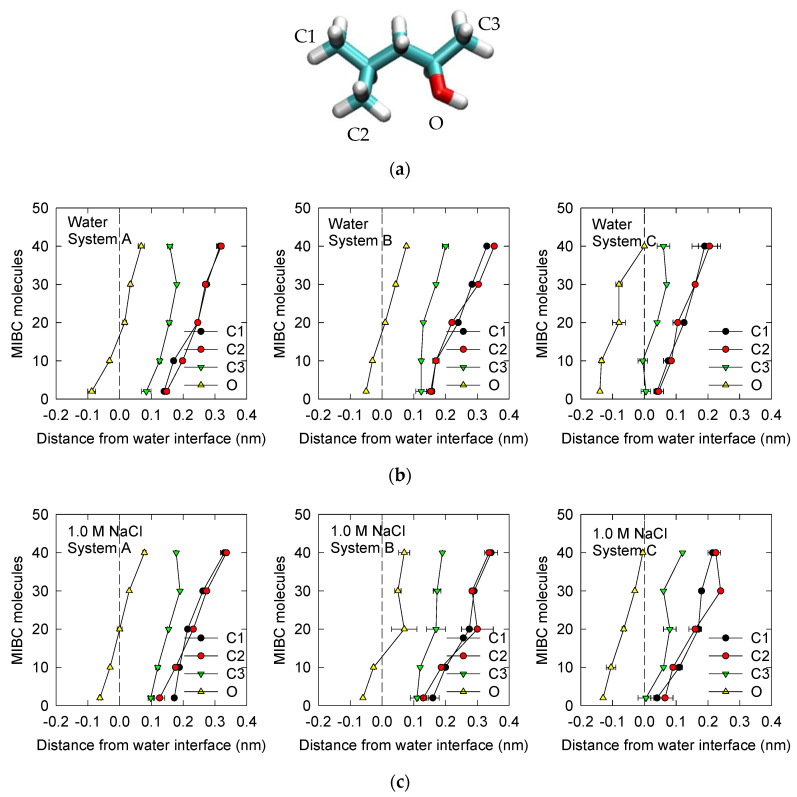
MIBC model indicating atom types, carbons (C1, C2 and C3) and oxygen (O) (**a**). Average position of MIBC molecules as represented by the constituent atoms with respect to the water liquid–vapor interface in water (**b**) and in 1 M NaCl (**c**) for the three systems of interest (Table 2). Negative distances correspond to liquid phase. Results correspond to the average of 10 simulations for each concentration of MIBC ranging from 2 to 40 molecules.

**Figure 4 polymers-14-01967-f004:**
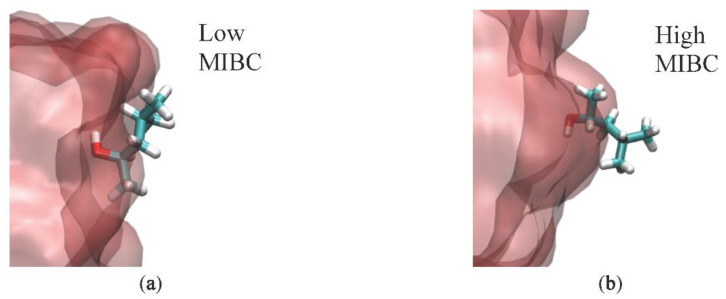
Spatial orientation of MIBC molecules at the liquid–water interfaces. (**a**) Parallel to the interface for low concentration and (**b**) orthogonal to the interface for high concentration. Both are in pure water and in 1 M NaCl solution.

**Figure 5 polymers-14-01967-f005:**
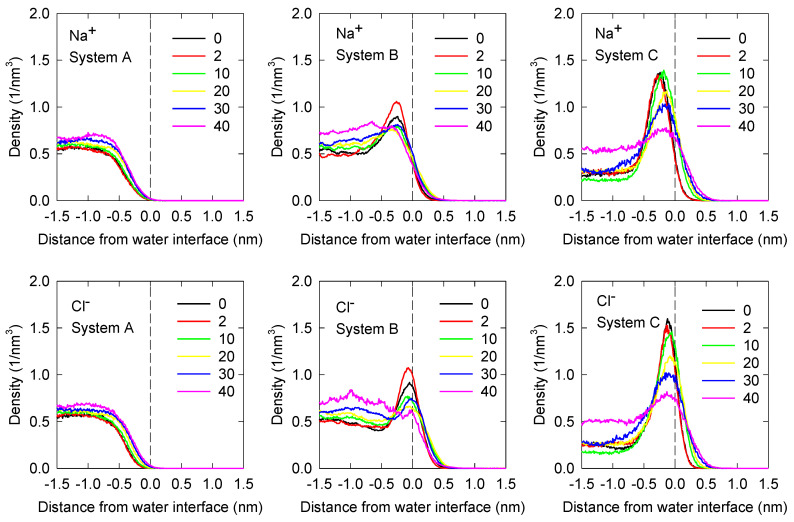
Density profiles of Na^+^ and Cl^−^ ions in the vicinity of the water liquid–vapor interface for the three systems of interest (Table 2). Negative distances correspond to liquid phase. The number of MIBC molecules is in the legends.

**Figure 6 polymers-14-01967-f006:**
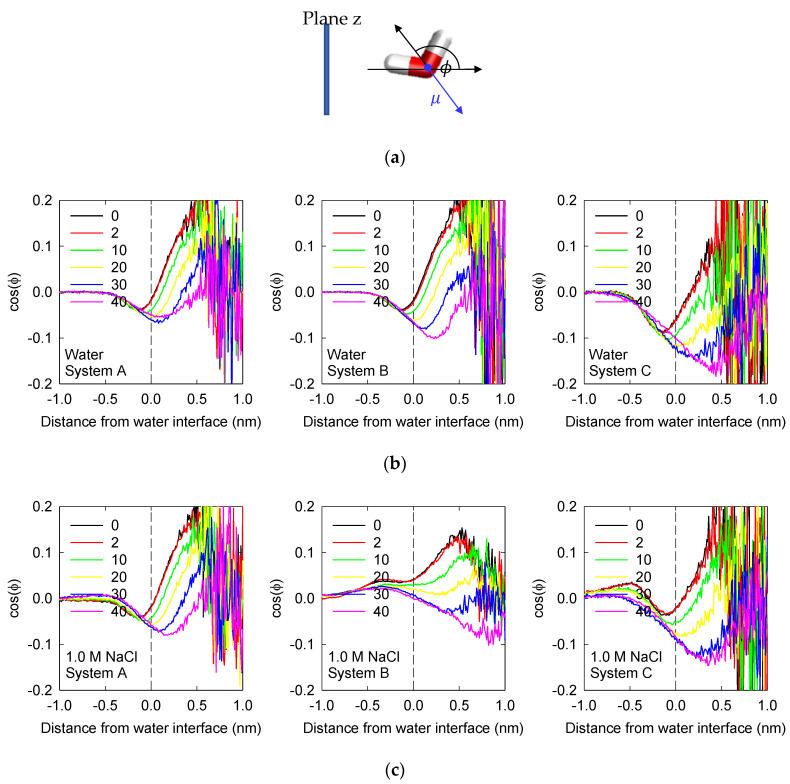
(**a**) Definition of water orientation with respect to the liquid–vapor interface. (**b**,**c**) Average water orientation represented by $\cos\left(\phi \right)$ as a function of the distance from the liquid–vapor interface (plane *z*) and the concentration of MIBC (number of molecules in the legends) for the three systems of interest (Table 2) in water and 1 M NaCl solution at 300 K. Negative distances correspond to liquid phase.

**Figure 7 polymers-14-01967-f007:**
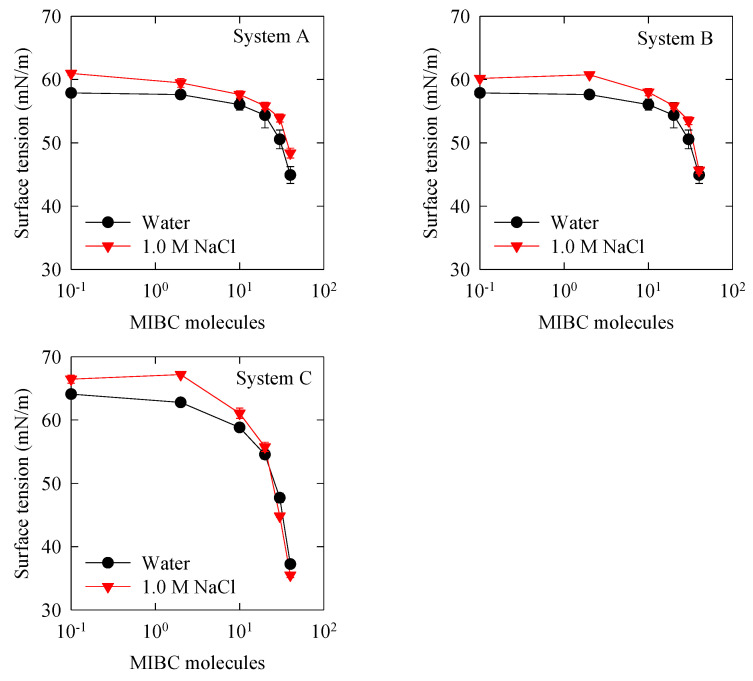
Surface tension versus concentration of MIBC in water and 1.0 M NaCl aqueous solution. Comparison of simulation results for the three systems of interest (Table 2).

**Table 1 polymers-14-01967-t001:** Partial charges for MIBC molecules.

Atom Type	Charge	Meaning
H	+0.06000	Aliplatic hydrogen
C1	−0.06000	Aliphatic carbon with 1 H
C2	−0.12000	Aliphatic carbon with 2 H
C3	−0.18000	Aliphatic carbon with 3 H
COH	+0.20500	Carbon bond with OH
OH	−0.68300	Hydroxide oxygen
HO	+0.41800	Hydroxide hydrogen

**Table 2 polymers-14-01967-t002:** Systems considered and force fields.

Components	Systems		
	A	B	C
Water model	SPC/E	SPC/E	SWM4-NDP
Ions model	Li-IOD-2015	Drude-2013	Drude-2013
MIBC model	Amber99sb	Amber99sb	Amber99sb

## Data Availability

The data presented in this study are available on request from authors J.H. Saavedra, R.G. Quezada and P.G. Toledo.

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
