# Peer review of "Species Surface Distribution and Surface Tension of Aqueous Solutions of MIBC and NaCl Using Molecular Dynamics Simulations"

_polymers, 2022, doi:10.3390/polym14101967_

Round 1

Reviewer 1 Report

This is an interesting and well-done simulation study of a surfactant (methyl isobutyl carbinol, MIBC) at the water/vapor interface, both with and without NaCl. The focus of this study is how polarizability, which is known to be important for ions a the water/air interface, affects the details of the interface.  I think this would be interest to the community and should be published. 

Reviewer 2 Report

The manuscript investigates the structure of air-liquid interfaces of aqueous solutions of MIBC-NaCL by means of MD simulation of all-atom force-fields. Most importantly, authors find the surface salvation of Cl- ions and the expulsion of non-polarisable Na+ ions from the interface to the bulk, which was also not able to be captured by nonpolarizable force-fields. Authors measure a number of properties. Of course properties, such as surface tension are not really well-reproduced by the force-fields, but this is well-known for these models.  Authors have obtained many interesting results, for a very interesting system. The manuscript is well-written, figures and literature are well taken care and the conclusions are supported by the results. Also, authors provide adequate explanations for the observed phenomena. I am happy to recommend publication of this manuscript as is.

Reviewer 3 Report

The study is well planned, structured and executed. I don't find any major issues in the manuscript. MDs and trajectory analysis are carefully and properly performed and I don't have any objections. I appreciate the effort to showcase the difference between polarazible and non-polarizable FFs.

As a recommendation, it would be beneficial for the Authors to think about X-ray and neutron reflectivity techniques. Having complementary experimental data would considerably increase scientific soundness. One can submit scientific proposals to large scale neutron/x-rays facilities around the world. This work would be a great background and motivation to obtain beam-time and experimentally study this and other air-liquid interfaces.

As a minor comment, I would ask Authors to consider a slight change to the title. It would be good to avoid abbreviations (MIBC) in the title. Not only in this work but in general. Please write it as methyl isobutyl carbinol.